

# The relation between human papillomavirus (HPV) and oropharyngeal cancer: a review

Chrystiano Campos Ferreira[1,2]

[1] Department of Medicine, Federal University of Rondonia, Porto Velho, Rondonia, Brazil
[2] Head and Neck Department, Barretos Cancer Hospital, Porto Velho, Rondonia, Brazil

## ABSTRACT

Oropharyngeal squamous cell carcinomas (OPSCC) represent a major public health challenge. In 2020, the international agency for research on cancer (IARC) recorded 98,421 cases of OPSCC worldwide. Over the past decade, the epidemiological profile of patients with OPSCC has shifted, mainly due to a change in etiological factors. Previously, alcohol and tobacco were considered the primary contributors, but the human papillomavirus (HPV) is now recognized as the leading cause of these tumors. This study aimed to conduct a literature review on the relationship between OPSCC and HPV for the general practitioner. The review examined the primary clinical differences between HPV$^+$ and HPV$^-$ OPSCC, their prognosis and treatment. In addition, the various HPV diagnostic methods were analyzed. Although there is a vast amount of literature on HPV, this review is unique in its ability to present the key information in an organized and accessible way and enables healthcare professionals to gain a better understanding of the relationship between HPV and oropharyngeal cancer. This, in turn, can contribute to the prevention of various cancers caused by the HPV virus, including oropharyngeal cancer.

## INTRODUCTION

Oropharynx squamous cell carcinomas (OPSCC) represent a major public health challenge. In 2020, the international agency for research on cancer (IARC) recorded 98,421 cases of OPSCC worldwide (*Ferlay et al., 2020*).

Over the past decade, the epidemiological profile of patients with OPSCC has shifted, mainly due to a change in etiological factors. Previously, alcohol and tobacco were considered the primary contributors, but the human papillomavirus (HPV) is now recognized as the leading cause of these tumors (*Wyss et al., 2013*; *Menezes et al., 2020*).

HPV-related OPSCC have different clinical presentation and prognoses when compared to non-HPV-related tumors. Thus, knowledge of the relationship between different epidemiological and clinical variables and HPV positivity in these tumors is crucial for better understanding and knowledge of this disease.

For oropharyngeal cancer, there is insufficient data in the literature, to date, to support the presence or absence of HPV/p16 as a relevant pathological variable in the diagnosis

Corresponding author
Chrystiano Campos Ferreira,
chrystiano_campos@yahoo.com.br

and prognosis of oral squamous cell carcinoma (*Wyss et al., 2013*; *Bouvard et al., 2009*). The relationship between HPV and oropharyngeal cancer is already well-established in the literature. However, there is still a need to better understand the clinical differences in the treatment of HPV$^+$ and HPV$^-$ OPSCC patients. The number of sexual partners is the primary determinant of anogenital HPV infection in both men and women (*Bouvard et al., 2009*). The prevalence of HPV in OPSCC varies according to the studied population, the methodology applied in its detection, and the decade in which the study was carried out. More current studies tend to show higher rates of HPV prevalence compared to older studies (*Kreimer et al., 2005*).

Developed countries generally exhibit higher incidences of HPV-related OPSCC than developing countries. For instance, *Chaturvedi et al. (2011)*, *Anantharaman et al. (2017)* and *D'Souza et al. (2007)* reported incidences of 44.1%, 59% and 72%, respectively, in US casuistry. *Näsman et al. (2009)* studied 98 patients with OPSCC in Sweden and found an HPV incidence of 79%. By contrast, populations from developing countries show lower incidences of HPV in OPSCC. For example, *López et al. (2014)* studied 91 patients in Brazil and found an incidence of 6.6% while *Anantharaman et al. (2017)* compared cases from the USA, Europe, and Brazil, finding an incidence of only 4.1% in 171 Brazilian cases. *Petito et al. (2017)* studied 82 cases in Brazil and found an incidence of 25.6%. A meta-analysis recently published in 2022, comprising data from 38 studies carried out in South American countries, revealed a prevalence of 24.31% of HPV in oropharyngeal tumors (*Oliveira et al., 2022*).

Although there is a vast amount of literature on HPV, this review is unique in its ability to present the key information (*Guadiana, Kavanagh & Squarize, 2021*) in an organized and accessible way and enables healthcare professionals to gain a better understanding of the relationship between HPV and oropharyngeal cancer. This, in turn, can contribute to the prevention of various cancers caused by the HPV virus, including oropharyngeal cancer.

This study aimed to conduct a literature review for the general practitioner analyzing the relationship between OPSCC and HPV. This review focuses on the main clinical differences between HPV$^+$ and HPV$^-$ OPSCC, their prognosis and treatment.

## SURVEY METHODOLOGY

In August 2022, a literature search was conducted using the MEDLINE/PubMed, and Google Scholar databases. The following keywords were used in the search ("HPV" OR "papillomavirus") AND ("oropharynx" OR "oropharyngeal") AND ("cancer" OR "tumor" OR "neoplasms") to identify articles reporting on HPV and its relation to oropharyngeal cancer. Articles published in English, Portuguese or Spanish were included, regardless of study design or publication date. The studies considered most relevant were selected for inclusion in this review and additional articles on HPV vaccination were also included to complement the analysis. Table 1 summarizes the key studies included in this review.

**Table 1 Characteristics of studies.**

| Study | Methods | Key findings |
| --- | --- | --- |
| *Anantharaman et al. (2017)* | Case-control study. | The proportion of OPSCC caused by HPV16 varies by geographic region with low proportions in Brazil, moderate proportions in Western Europe, and the majority in the U.S. being HPV16-positive. |
| *Ang et al. (2010)* | Retrospective study. | HPV status is a strong and independent prognostic factor for survival among patients with OPSCC. |
| *Carpén et al. (2018)* | Retrospective study. | OPSCC can be dichotomized in two distinct disease entities as defined by HPV status. |
| *Chaturvedi et al. (2011)* | Cross-sectional study. | Increases in the population-level incidence and survival of oropharyngeal cancers in the United States since 1984 are caused by HPV infection. |
| *Chaturvedi et al. (2018)* | Cross-sectional study. | HPV vaccination was associated with reduction in vaccine-type oral HPV prevalence among young US adults. |
| *Chera et al. (2019)* | Phase II Trial. | Clinical outcomes with a de-intensified chemoradiotherapy regimen of 60 Gy intensity-modulated radiotherapy with concurrent low-dose cisplatin are favorable in patients with human papillomavirus–associated OPSCC. |
| *de Ferreira et al. (2021)* | Cross-sectional study. | Drinkers and current smokers were less likely to be p16+. |
| *D'Souza et al. (2007)* | Case-control study | Oral HPV infection is strongly associated with oropharyngeal cancer among subjects with or without the established risk factors of tobacco and alcohol use. |
| *D'Souza et al. (2009)* | Cross-sectional study. | Oral sex and open-mouthed kissing are associated with the development of oral HPV infection. |
| *D'Souza et al. (2014)* | Multicenter prospective study. | Oral HPV16 DNA is commonly detected among patients with HPV-OPC at diagnosis, but not among their partners. |
| *Fakhry et al. (2008)* | Prospective study. | HPV status is associated with therapeutic response and survival on OPSCC patients. |
| *Gillison et al. (2008)* | Case-control study. | HPV$^+$ HNSCCs and HPV$^-$ head and neck cancer have different risk factor profiles. |
| *Gillison et al. (2012)* | Cross-sectional study. | Among men and women aged 14 to 69 years in the United States, the overall prevalence of oral HPV infection was 6.9%. |
| *Kreimer et al. (2013)* | Cohort study. | HPV16 E6 seropositivity was present more than 10 years before diagnosis of oropharyngeal cancers. |
| *Lassen et al. (2009)* | Cohort study. | Expression of p16 has a major impact on treatment response and survival in patients with head and neck cancer treated with conventional radiotherapy. |
| *López et al. (2014)* | Cohort study. | Seropositivity for HPV16 E6 antibodies was correlated with improved Head and neck cancer survival and oropharyngeal cancer. |
| *Menezes et al. (2020)* | Population-based study | Emerging risk for HPV-related OPSCC in young people, which supports prophylactic HPV vaccination in this group. |
| *Näsman et al. (2009)* | Retrospective study. | A increase in tonsillar cancer mainly due to HPV infection. |
| *Petito et al. (2017)* | Retrospective study. | |
| *Posner et al. (2011)* | | HPV$^+$ OPSCC has a different biology compared with HPV-. |
| *Raman et al. (2019)* | Retrospective study. | The increased awareness and complexity of treatment decisions related to OPSCC may affect times to diagnosis and treatment initiation in patients with HPV-positive disease. |

# LITERATURE REVIEW

## Relationship between HPV and head and neck cancer

HPV represents the most common sexually transmitted infection caused by a virus in the world (*Muñoz et al., 2006*; *Bouvard et al., 2009*). HPV infections are transmitted primarily through direct skin-to-skin or skin-to-mucosa contact and the number of sexual partners

is the main determinant of anogenital HPV infection in both men and women (*Bouvard et al., 2009*).

More than 200 types of HPV are known, and the eight most common types of HPV worldwide are HPV 16, 18, 31, 33, 35, 45, 52, and 58. Type 16 is the most common of all, followed by type 18 in various parts of the world (*Muñoz et al., 2006*; *Bouvard et al., 2009*).

The prevalence of HPV in the oropharynx and oral cavity varies widely across different populations. A study by *Gillison et al. (2012)* investigated the presence of the HPV virus in the oral mucosa of 5,579 individuals and showed a prevalence of 6.9%, with higher prevalence in men than in women. Another study, by *D'Souza et al. (2014)*, looked for the presence of HPV through mouthwashes of patients with oropharyngeal cancer associated with HPV. The presence of HPV was detected in 65% of the cases. Among the partners of the patients, an incidence of HPV of 4% was obtained, suggesting that most partners can clear the virus when exposed.

In a meta-analysis performed by *Kreimer et al. (2005)*, positivity for HPV in tumor biopsies was found in 35.6% of cases of OPSCC, with type 16 being the most common type, present in 30.9% (86.8% of positive cases). Type 18 was found in only 1.0% of tumors and type six was found in 2.5% of tumors.

Smoking and alcohol consumption are the most important risk factors for head and neck cancer. However, many patients with OPSCC do not have any of the traditional risk factors. Many studies have correlated HPV as an etiological agent in many of these patients (*Vokes, Agrawal & Seiwert, 2015*).

HPV is present in head and neck mucosal regions at a rate of 7% in healthy individuals (*Gillison et al., 2012*). Most HPV infections typically end spontaneously within 12 months (*Plummer et al., 2007*; *Rodríguez et al., 2008*) and the time between exposure to HPV and the development of oropharyngeal cancer varies but is likely to exceed 10 years (*Kreimer et al., 2013*; *Robbins et al., 2022*).

In 2007, the international agency for research on cancer (IARC) considered HPV as a known carcinogenic for the oral cavity and oropharynx. However, the evidence linking HPV to cancer in the larynx was found to be limited (*Bouvard et al., 2009*).

Several studies point to HPV type 16 as the etiological agent in many patients with head and neck cancer and HPV 16 is present in 90% of cases of HPV[+] (*D'Souza et al., 2007*; *Bouvard et al., 2009*; *Gillison et al., 2012*). *Mehanna et al. (2013)* found an increase in the number of HPV positivity cases in oropharyngeal cancers over the past decades, but this has not been accompanied by an overall increase in the number of OPSCC cases in the general population.

One factor related to the increase in cases of oropharyngeal cancer is the change in the population's sexual behavior. According to *D'Souza et al. (2009)*, engaging in kissing and oral sex are considered risk factors for HPV infection in the oral cavity and oropharynx. Furthermore, having more than six partners who have engaged in oral sex was identified as a risk factor for OPSCC.

**Table 2 Clinical characteristics of patients HPV+ and HPV−.**

| | HPV+ | HPV− |
|---|---|---|
| Age | Younger patients | Older patients |
| Sex | More frequent in men | Frequent in men |
| Sexual behavior | Oral sex, high number of sexual partners | Variable |
| Primary tumor and metastasis | Small primary tumors in contrast to large metastases | Variable |
| Alcohol and tobacco | Frequent patients who do not smoke or drink | Frequent patients who smoke or drink |
| Prognosis | Better survival outcomes and better response to treatment | Worse |

## Clinical presentation

Clinical symptoms of patients with OPSCC may include dysphagia, odynophagia, neck mass or otalgia. Compared with non-HPV+, these are more likely to have cervical lymph node enlargement at presentation (*Raman et al., 2019*). Clinical presentation of HPV+ and HPV− individuals are listed in Table 2.

Patients with HPV-positive OPSCC display distinct characteristics compared to those who are HPV-negative. Typically, HPV-positive patients are younger and predominantly male, as evidenced by the ICON-S study, which found that 84% of HPV-positive patients were male, compared to only 76% of HPV-negative patients (*O'Sullivan et al., 2016*). Additionally, HPV-positive patients tend to present with initial tumors (T1 or T2) that are associated with advanced cervical metastasis (N2 or N3), according to research by *Chaturvedi et al. (2011)* and *O'Sullivan et al. (2016)*.

Another distinguishing characteristic of HPV-positive OPSCC is its more frequent location in the tonsils or at the base of the tongue. According to *Posner et al. (2011)*, who studied 111 patients with OPSCC, HPV-positive patients tend to be younger (with a median age of 54 compared to 58 for HPV-negative patients) and are more likely to be diagnosed with early-stage tumors. Specifically, 49% of HPV-positive patients had T1 or T2 tumors, compared to only 20% of HPV-negative patients (*Posner et al., 2011*).

In a study carried out by *Gillison et al. (2012)* at Johns Hopkins Hospital, 240 patients diagnosed with OPSCC were divided into two groups based on their HPV status, determined by *in situ* hybridization and negative group. Of the 240 cases, 92 were HPV+. The study found that compared with patients with HPV negative tumors, patients with HPV+ were younger (mean age 54 years compared to 58 years), with a higher predominance of caucasian, more years of education, higher family income and more likely to be married.

The ICON-S study found that the mean age of HPV-positive OPSCC patients was 57 years, which is younger than the mean age of 61 years for HPV-negative patients. This difference in age may be attributed to a decrease in smoking and alcohol use among HPV-positive patients, which is associated with better treatment tolerance and improved prognosis (*O'Sullivan et al., 2016*).

In general, HPV$^+$ patients have lower rates of smoking and alcoholism when compared to HPV$^-$ patients and generally report a greater number of sexual partners and/or frequency of oral sex (*Gillison et al., 2008*; *D'Souza et al., 2009*).

Furthermore, tumors associated with HPV infection are often detected through asymptomatic cervical metastasis. Patients with HPV-positive OPSCC may not present with complaints related to the primary tumor. Instead, the most common presentation is the detection of cervical metastasis without any associated symptoms (*Carpén et al., 2018*).

## Propedeutics and diagnosis

When approaching a patient with oropharyngeal squamous cell carcinoma (OPSCC), a comprehensive medical history is crucial to identify risk factors such as smoking, alcohol consumption, and sexual behavior. A physical examination is also necessary, including inspection of the oropharynx, palpation of all lymph node levels in the neck, and nasolaryngoscopy or laryngoscopy to evaluate the extent of the tumor. Biopsy of the suspicious lesion in the oropharynx and fine needle aspiration biopsy of cervical masses should be performed. For accurate staging, a CT scan of the neck and thorax is required. Additionally, it is important to determine whether the tumor is positive for human papillomavirus (HPV). In general, immunohistochemistry for p16 is performed on the biopsy to detect the presence of HPV (*Alabi & O'Neill, 2020*).

HPV can be diagnosed using two methods: detection of viral DNA or RNA and detection of cellular markers. However, the most common method used for OPSCC is immunohistochemical detection of p16, an oncoprotein related to HPV activity (*Lassen et al., 2009*; *Ferreira et al., 2023*; *de Ferreira et al., 2021*). The advantages of using p16 overexpression research by immunohistochemistry include its practicality, simplicity, and low cost (*Robinson et al., 2012*; *Fakhry et al., 2018*). A study of 316 OPSCC patients who underwent p16 and HPV research by *in situ* hybridization found that 10% of p16-positive patients were HPV-negative, while 7% of p16-negative patients were HPV-positive (*Fakhry et al., 2018*).

Several major guidelines, such as the national comprehensive cancer network (NCCN), recommend the clinical use of p16 immunohistochemistry to characterize cases as HPV-positive and HPV-negative cases (*Robinson et al., 2012*; *Fakhry et al., 2018*). The 8th edition of the AJCC staging system also incorporates HPV status, leading to changes in both the staging and prognosis of OPSCC (*Robinson et al., 2012*; *Fakhry et al., 2018*).

It is therefore recommended that all patients undergoing biopsy of tumors in the oropharynx region be asked to carry out a p16 immunohistochemistry test in addition to an anatomopathological examination of the specimen. This will help to ensure accurate diagnosis and appropriate treatment selection for OPSCC patients.

## Prognosis

HPV-positive OPSCC has a better prognosis compared to HPV-negative OPSCC (*Fakhry et al., 2008*; *Lassen et al., 2009*; *Schiller & Lowy, 2012*). The 8th edition of the TNM staging system proposed a new staging for OPSCC, with distinct staging criteria for p16-positive and p16-negative tumors. This change in staging reflects the significant difference in

prognosis between these two subtypes of OPSCC. In the 7th edition, only T1 or T2 N0, M0 were considered early stage, regardless of HPV status. However, in the 8th edition, HPV-negative tumors continue to be considered early stage T1 or T2 N0, M0, but positive HPV T3 and N2 were also considered initial (*Robinson et al., 2012*; *Fakhry et al., 2018*).

*Ang et al. (2010)* conducted a study on 323 patients with stage III and IV OPSCC, using both hybridization for viral DNA and immuno-histochemical analysis of p16. The results indicated that 206 patients (63.8%) were HPV positive. The 3-year survival rate was 82.4% in the HPV$^+$ group and 57.1% in the HPV-negative group, while the 3-year disease-free rate was 73.7% in the HPV$^+$ group and 43.4% in the HPV$^-$ group. Furthermore, the study revealed that the 3-year survival rate was 83.6% for the HPV$^+$ group and 51.3% for the HPV$^-$ group, and the 3-year disease-free rate was 74% for the HPV$^+$ group and 38.4% for the HPV$^-$ group. The study identified an intermediate-risk group of patients with OPSCC who were HPV$^+$ and had more than 10 pack-years of tobacco smoking, which had an intermediate prognosis (*Ang et al., 2010*).

In a study conducted by *Fakhry et al. (2008)*, 96 patients with stage III or IV cancer of the oropharynx and larynx were evaluated for the presence of HPV DNA using PCR and *in situ* hybridization. HPV DNA was positive in 38 patients (40%). The study showed that disease-free survival in HPV$^+$ patients was 91% in the first year and 86% in the second year, while it was 69% in the first year and 53% in the second year for HPV- patients. Patients with HPV$^+$ tumors had higher response rates after induction chemotherapy (82% *vs* 55%) and after radiochemotherapy treatment (84% *vs* 57%) compared to those with HPV$^-$ tumors. Overall 2-year survival was 95% for HPV$^+$ patients compared to 62% for HPV$^-$ patients.

## Vaccination against HPV

The risk behavior most associated with HPV-related OPSCC is oral sex. Thus, prevention should promote actions such as the use of condoms and a reduction in the number of sexual partners.

Prevention for HPV-related OPSCC includes vaccination against HPV, which is recommended for people aged 9 to 26 years old. Currently, there are three types of HPV vaccines approved by the FDA: Gardasil-9, approved in 2014 (which prevents types 6, 11, 16, 18, 31, 33, 45, 52, and 58), Cervarix, approved in 2009 (which prevents types 16 and 18), and Gardasil, approved in 2006 (which prevents types 6, 11, 16 and 18) (*Krajden et al., 2011*; *Romanowski et al., 2011*; *Schiller & Lowy, 2012*). The recommended age for vaccination is between 9 and 26 years, with Cervarix indicated for girls and Gardasil and Gardasil-9 indicated for both boys and girls (see Table 3). While vaccination is not recommended for individuals over the age of 26, some adults aged 27 through 45 years may still receive the HPV vaccine based on a discussion with their doctor, especially if they have not been previously vaccinated (*Zhang, Fakhry & D'Souza, 2021*). While all three vaccines have been shown to be effective in preventing anogenital infections and malignancies caused by the HPV types included in the vaccines, there is limited evidence to suggest that they can decrease the incidence of head and neck tumors (*Schiller & Lowy, 2012*).

**Table 3 FDA-approved types of vaccines for HPV.**

| Vaccine | HPV type | Sex and age |
|---|---|---|
| Cervarix (bivalent) (GlaxoSmithKline) | 16 and 18 | Female (9–25 years)[*] |
| Gardasil (quadrivalent) (Merck) | 6, 11, 16, and 18 | Male and female (9–26 years)[*] |
| Gardasil-9 (9-valent) (Merck) | 6, 11, 16, 18, 31, 33, 45, 52, 58 | Male and female (9–26 years)[*] |

Note:
[*] Some adults ages 27 through 45 years might get the HPV vaccine based on discussion with their doctor.

The HPV vaccine has demonstrated a reduction in the prevalence of oral HPV infection. In a study of 2,627 participants aged 18 to 33 years, both men and women were analyzed for the presence or absence of HPV in mouthwashes 4 years after vaccination. Of the participants, 18.3% received one or more doses of the vaccine. The study observed a prevalence of HPV types 16, 18, 6 or 11 in the oral cavity of 0.11 in vaccinated individuals *vs* 1.61% in non-vaccinated individuals (*Gillison et al., 2008*). It has been reported that the effects of the HPV vaccine on reducing oral HPV infections may persist for over 10 years after vaccination (*Rowhani-Rahbar et al., 2009*).

Another important point is the potential for health professionals to be contaminated with HPV during procedures that emit aerosols, such as excision or electrocoagulation of HPV-related lesions. Although the level risk is still uncertain, evidence suggest a real risk may exist (*Harrison & Huh, 2020*).

## Treatment of oropharyngeal cancer

The treatment of oropharyngeal cancer requires a multidisciplinary team. Treatment options may include radiotherapy, surgery and/or chemotherapy and the choice of treatment depends on the clinical stage of the tumor and the clinical condition of the patient. Typically, radiotherapy or surgery are indicated for early-stage T1 and T2 tumors without cervical metastasis, and radiotherapy associated with chemotherapy is recommended for more advanced cases.

Although assessing the patient's HPV status is important for staging and predicting prognosis, there is still insufficient support to carry out different treatment options. Therefore, the treatment for OPSCC associated with HPV should be the same for HPV-negative cases except in clinical trials.

Some studies have investigated the de-escalation of RT and/or chemotherapy treatment showing encouraging results in phase II non-randomized studies (*Marur et al., 2017*; *Chera et al., 2019*). Due to the favorable prognosis of patients after standard therapy, further research is needed to better evaluate the efficacy and safety of treatment de-escalation as a standard practice.

## CONCLUSIONS

This review described the relationship between the HPV virus and OPSCC. The various HPV diagnostic methods were analyzed and an explanation was given for p16 being used as a marker for HPV infection. The clinical differences between HPV[+] and HPV[−] OPSCC were also discussed, including their impact on prognosis and treatment. This information

is crucial as OPSCC is a significant public health issue and preventive measures can be taken in most cases if the virus is identified early.

While there is a wealth of literature on HPV, this review presents key information in an organized and accessible way that can help increase awareness and understanding among health professionals on the relationship between HPV and oropharyngeal cancer. This can lead to a reduction in the incidence of HPV-related cancers, including OPSCC, and improve patient outcomes.

## ACKNOWLEDGEMENTS

We thank the Barretos Cancer Hospital.

### Funding
The author received no funding for this work.

### Competing Interests
The author declares that he has no competing interests.

### Author Contributions
- Chrystiano Campos Ferreira conceived and designed the experiments, performed the experiments, analyzed the data, prepared figures and/or tables, authored or reviewed drafts of the article, and approved the final draft.

### Data Availability
This is a literature review.

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
