# Peer review of "The relation between human papillomavirus (HPV) and oropharyngeal cancer: a review"

_PeerJ, doi:10.7717/peerj.15568_

## Round 0.1 · original submission · Major Revisions

Reviewers raised several points that need to be addressed. Please pay attention, especially to the methods used, grammar, and improve the abstract

·

Basic reporting

The author shares a literature search on OPSCC in relation to HPV infections.

The topic is highly relevant and the authors efforts in completing the manuscript are acknowledged.

However, some style comments - the paper is presented by a single author yet numerous time "we" and "our" is mentioned, that needs to be changed.
There are many places where important statements are made that are not backed up by literature. Among these but not limited: Line 118, 119, and 133. The author is encouraged to carefully go through other areas of the text and add relevant references to important claims.

Overall the study misses a relevant summary for example in form of an overview table that contains key observations of the studies discussed.
The order of the sections in not ideal especially the sections line 155, line 173 and line 204 are not intuitively ordered nor do they reach enough depth to justify the headline, it appears more as an extended introduction.

The author has also chosen to copy and paste text sections into the abstract, which while slightly irritating to the reader also doesn't make for the most informative abstract, the parts of the results are actually methods and the conclusions do not include any conclusions from the review.

The conclusions and assumptions about p16 will need to be reworded as this is very unclear.

Line 134 - natural history might not be a good fit here
Line 75 - there is not discussion or definition of "biological behavior" this may not be what the author meant to say.

Experimental design

it is unclear for the author used the described serach terms all together, or seperate, or partially together, more clarity is needed, also please add the date of the oldest and the newest study.

As mentioned above, consider a summary table of all the included studies with key elements.

Validity of the findings

The conclusions at this point fall short and will need to be reworded for final approval.

Reviewer 2 ·

Basic reporting

Please be more specific when reporting figures. eg in line 127, you mention "positivity for HPV" without any indication of what is being tested (primary tumors, LN biopsies/FNAs, blood). I assume this refers to tumor biopsies (either primary tumors or lymph nodes, but it would be best to clarify.

The authors fail to mention the identification of an intermediate risk group of patients with OPSCC: HPV+ OPSCC in patients with >10 pack years (Ang 2010). This would best be included in the the paragraph that begins on line 259.

The authors need to update their HPV vaccination recommended age range as it has been expanded to patients up to the age of 45 for 1 or more of the available vaccines.

Experimental design

why search "oral cancer". It would have been better to subsitute this search for "oropharyngeal cancer" or "oropharynx cancer". The authors claim a link between HPV and cancer of the oral cavity, however, I believe this is an overstatement. At best, the link is unclear/unproven to date and does not belong in a review of this nature.

Validity of the findings

see above

Additional comments

This review is relatively cursory and once revised may be an appropriate manuscript for the general practitioner. It requires rigorous grammatical and punctuation reviews prior to acceptance as there are several incomplete and run on sentences, missing punctuation, etc.

---

## Round 0.2 · Minor Revisions

There are still issues raised by reviewers

Reviewer 2 ·

Basic reporting

The authors made improvements throughout the manuscript and I believe it is a relatively concise review appropriate for the general practitioner. I recommend changing "initial" (lines244-246) to "early stage" for better clarity.

I would also recommend the authors clarify that there is insufficient data in the literature, to date, to support the presence or absence of HPV/p16 as a relevant pathological variable in the diagnosis and prognosis of oral squamous cell carcinoma - as opposed to the oropharyngeal subsites of disease.

Experimental design

.

Validity of the findings

.

Additional comments

.

---

## Round 0.3 · accepted · Accept

The manuscript is now more robust.

Reviewer 2 ·

Basic reporting

Adequate changes have been made to merit publication.

Experimental design

.

Validity of the findings

.

Additional comments

.